# DNA transposon activity is associated with increased mutation rates in genes of rice and other grasses

Thomas Wicker[1], Yeisoo Yu[2,†], Georg Haberer[3], Klaus F.X. Mayer[3], Pradeep Reddy Marri[4], Steve Rounsley[4,†], Mingsheng Chen[5], Andrea Zuccolo[6], Olivier Panaud[7], Rod A. Wing[2,8,9] & Stefan Roffler[1]

DNA (class 2) transposons are mobile genetic elements which move within their 'host' genome through excising and re-inserting elsewhere. Although the rice genome contains tens of thousands of such elements, their actual role in evolution is still unclear. Analysing over 650 transposon polymorphisms in the rice species *Oryza sativa* and *Oryza glaberrima*, we find that DNA repair following transposon excisions is associated with an increased number of mutations in the sequences neighbouring the transposon. Indeed, the 3,000 bp flanking the excised transposons can contain over 10 times more mutations than the genome-wide average. Since DNA transposons preferably insert near genes, this is correlated with increases in mutation rates in coding sequences and regulatory regions. Most importantly, we find this phenomenon also in maize, wheat and barley. Thus, these findings suggest that DNA transposon activity is a major evolutionary force in grasses which provide the basis of most food consumed by humankind.

[1] Department of Plant and Microbial Biology, University of Zurich, 8008 Zurich, Switzerland. [2] Arizona Genomics Institute, School of Plant Sciences, University of Arizona, Tucson, Arizona 85721, USA. [3] Plant Genome and Systems Biology, Helmholtz Center Munich, 85764 Neuherberg, Germany. [4] Dow AgroSciences, Indianapolis, Indiana 46268, USA. [5] State Key Laboratory of Plant Genomics, Institute of Genetics and Developmental Biology, Chinese Academy of Sciences, Chaoyang District, Beijing 100101 China. [6] Institute of Life Sciences, Scuola Superiore Sant'Anna, 56127 Pisa, Italy. [7] Laboratoire Génome et Développement des Plantes, UMR5096 UPVD/CNRS, Université de Perpignan Via Domitia, 66860 Perpignan, France. [8] International Rice Research Institute, Los Baños, 4031 Laguna, Philippines. [9] Department of Ecology and Evolutionary Biology, University of Arizona, Tucson, Arizona 85721, USA. † Present address: Phyzen Genomics Institute, Phyzen Inc., Seoul 151–836, South Korea (Y.Y); Genus plc, DeForest, WI 53532, USA. (S.R). Correspondence and requests for materials should be addressed to T.W. (email: wicker@botinst.uzh.ch).

The grass (Poaceae) family contains over 10,000 species and includes the most important agricultural crops such as rice, maize, wheat and barley. Grasses evolved from a common ancestor $\sim 70$ Myr ago[1]. One unique characteristic of grass genomes is that they contain enormous numbers of DNA (class 2) transposons. For example, the superfamilies *DTT_Mariner* and *DTH_Harbinger* alone are present in at least 40,000 copies in grass genomes[2,3]. Interestingly, the vast majority of DNA transposons in grasses are non-autonomous, meaning that they rely for their transposition on enzymes encoded by a small number of 'mother' elements elsewhere in the genome[3,4]. Furthermore, these small non-autonomous transposons were reported to preferably insert near genes[3,5,6]. But despite the high abundance of DNA transposons in grass genomes, little is known about their level of activity and their overall impact on genome evolution. This was mostly due to the lack of suitable data sets for comparative analyses. With the recent sequencing of 11 rice genomes in the framework of the Oryza Map Alignment Project (OMAP[7]), data sets for such studies became available. In this study, we compared the two rice species *Oryza sativa* and *Oryza glaberrima* which diverged $\sim 600,000$ years ago[8]. These two species are closely enough related to allow reliable alignment of most of the genomes and yet distant enough to have numerous transposable element (TE) polymorphisms[9,10].

DNA transposons have the curious ability to move in the genome by inserting into and excising from genomic DNA. When they excise from the genome, they leave double-strand breaks (DSBs) that have to be repaired by the cell. Previous studies have shown that, this can lead to deletions and/or insertions of 'filler' sequences at the site of the DSB[4,9,11], depending on the repair pathway. Sometimes, re-arrangements at the excision site can be so extensive that excisions are difficult to identify[9,11] (Supplementary Note 1). Thus, the previous studies have established that transposons leave a variety of 'scar' patterns at the site of excision. However, DSB repair is a highly complex process that involves multiple enzymes and, in some pathways, single-stranded DNA intermediates[12-22] (Supplementary Note 1). Considering these complex processes, we wanted to study if and to what degree DNA transposons excisions also affect the sequences surrounding the excision site and whether they have an impact on the evolution of genes. Our data suggest that transposon excisions invoke DNA repair mechanisms that lead to high numbers of mutations around the excisions sites. The preference of DNA transposons to insert near genes in grasses therefore accelerates evolution of genes and coding regions.

## Results

**Transposon excisions are flanked by numerous mutations.** For our analysis, we annotated 27,641 DNA transposons in the *O. sativa* genome; the majority of them belong to the *DTT_Mariner* and *DTH_Harbinger* superfamilies. Overall, they show a strong preference to insert close to transcription start and end points of genes (Supplementary Fig. 1). This is in agreement with previous findings that showed a preference of these elements to insert near genes[3,5,6] (Supplementary Note 2). To identify DNA transposon polymorphisms, we compared the annotated transposon loci with their orthologs in *O. glaberrima*. We manually screened over 2,000 potential polymorphisms and classified 482 as insertions and 158 as excisions (Table 1; Supplementary Tables 1 and 2; Supplementary Note 3). The polymorphic transposons belong to five different superfamilies of which *DTT_Mariner* and *DTH_Harbinger* elements comprise the majority (Table 1). Here, we made particular efforts to ensure that indeed orthologous loci were compared (Methods, Supplementary Fig. 2 Supplementary Note 4).

Interestingly, we found that excisions often go along with the introduction of numerous nucleotide substitutions and small insertions and deletions (InDels) in sequences flanking the transposons, with some flanking regions containing over 10 times more mutations than the genome on average (example in Fig. 1). To quantify this effect, we analysed the 12 kb flanking each polymorphic transposon and added up all nucleotide substitutions and (InDels) relative to the transposon insertion/excision site. The resulting plot shows that the overall frequency of nucleotide substitutions and InDels increases in an exponential manner towards the TE excision site to at least four-fold on average, compared with randomly picked genomic sequences (Fig. 2). Numbers of nucleotide substitutions and InDels are increased up to a distance of 3 kb from the excision point (Fig. 2). In contrast, transposon insertion sites have many fewer mutations in flanking regions, showing only a small increase in nucleotide substitution frequency in their neighbourhood (Figs 2; 3, see below).

**A proposed model for how transposon excisions induce mutations.** Considering findings on DSB repair from yeast[12-18] and *Arabidopsis*[19-22], we propose a molecular mechanism that explains the high numbers of mutations flanking transposon excisions in rice (Fig. 2c): in the first step, the transposons excise from the genome, leaving a DSB for the cell to repair. After transposon excision, 3' overhangs are produced by exonucleases (Fig. 2c, step 1). The 3' overhangs then anneal using micro-homologies of a few bp (Fig. 2c, step 2), or through an intermediate generated by invasion of a foreign strand (Supplementary Note 1). Subsequently, the single-stranded DNA segments are used as templates for the synthesis of a new second strand, which is the step that introduces numerous mutations (Fig. 2c, step 3). We propose that DNA replication is analogous to that described for DSB-induced replication in yeast[13]. Here, mutations are introduced by translesion synthesis, possibly involving a homologue of DNA polymerase zeta (which is involved in error-prone DNA repair in yeast[13]) and by a DSB-induced replication complex that has deficiencies in DNA polymerase delta fidelity and mismatch repair, analogous to that described in yeast[14]. Possibly, Rev1 polymerase also contributes to erroneous DNA repair[22]. The end product of the repair process are sequence segments flanking the transposon excision which are riddled with nucleotide substitutions and small InDels (Fig. 1 and Fig. 2c, step 4). The length of the segment containing the mutations depends on the size of the 3' overhang produced in the initial repair step. In yeast, these overhangs can be several kb in size[12,13,15], and this is expected to be similar in plants, due to the high conservation of DSB repair pathways[18]. Indeed, our data support this notion, since the observed average nucleotide substitution frequency levels off $\sim 3$ kb away from the excision site (Fig. 2a,b).

**Table 1 | Transposition events identified and manually curated in the comparison of the two rice species *O. sativa* and *O. glaberrima*.**

| Superfamily | Insertions | Excisions |
|---|---|---|
| *DTH_Harbinger* | 241 | 71 |
| *DTT_Mariner* | 137 | 64 |
| *DTM_Mutator* | 77 | 20 |
| *DTA_hAT* | 23 | 1 |
| *DTC_CACTA* | 4 | 2 |
| Total | 482 | 158 |

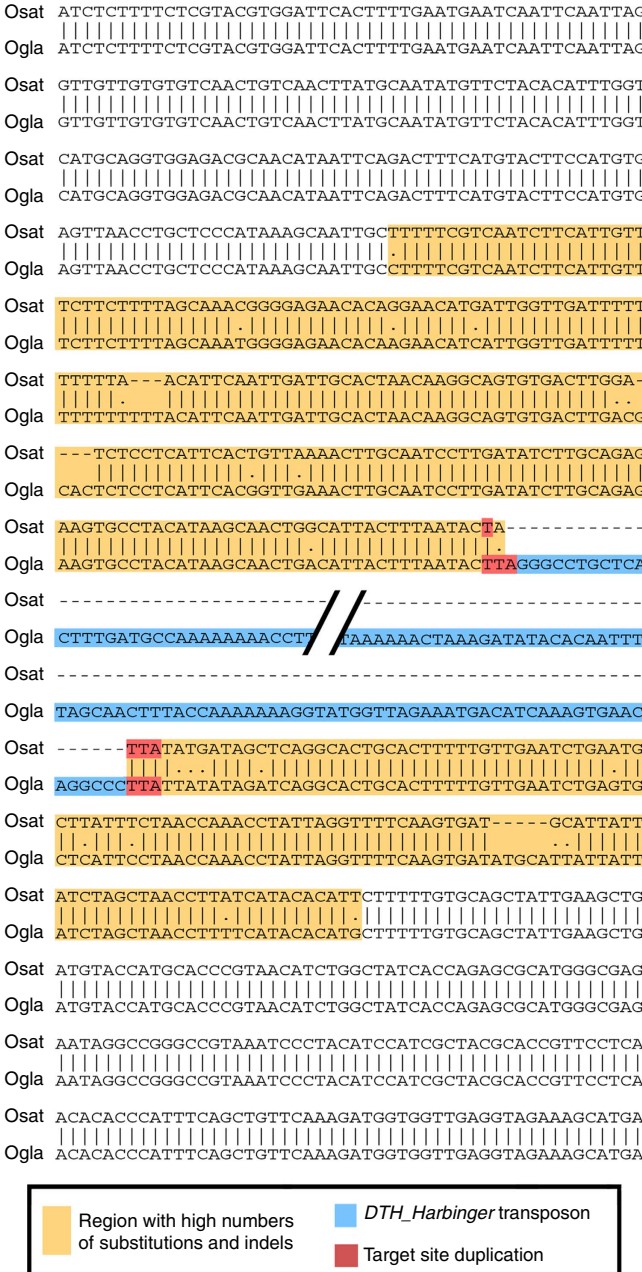

**Figure 1 | Example for a DNA transposon excision with numerous nucleotide substitutions in its flanking region.** A *DTH_Harbinger* transposon was excised from the genome of *O. sativa* (Osat) while it remained present in *O. glaberrima* (Ogla). In this particular event, the transposon was excised almost perfectly, only losing 2 bp of the target site duplication and replacing one of them with a mismatching base. The 211 bp upstream and 120 bp downstream of the excision contain 25 mutations (InDels >1 bp are counted as one mutation), resulting in <93% sequence identity and thus making the mutation rate over 15 times higher than for the genome overall. Outside of the region with the mutations, *O. sativa* and *O. glaberrima* sequences are identical, reflecting the overall genome-wide sequence conservation of ~99.5%. The segments shown correspond to *O. sativa* chromosome 1 position 23,814,561–23,815,081 and *O. glaberrima* chromosome 1 position 16,579,166–16,580,116.

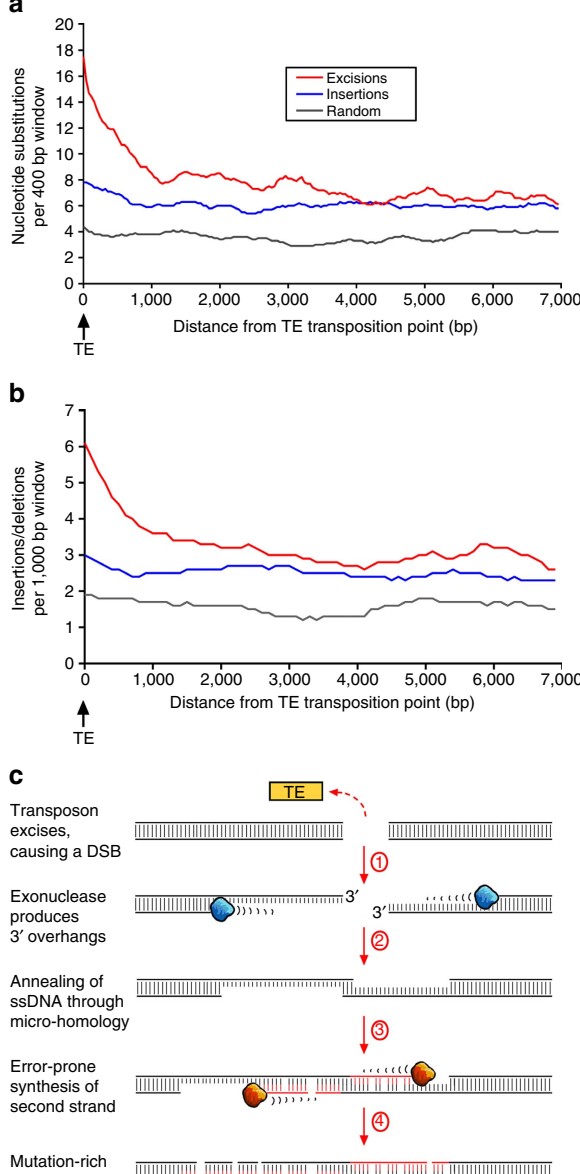

**Figure 2 | Mutations in sequences flanking transposon insertion and excision sites.** (**a**) Frequencies of nucleotide substitutions relative to transposon insertion/excision sites in rice. For the plot, 438 sequence alignments carrying transposon insertions (blue line) and 206 alignments carrying excisions (red line) were compiled. As control, 340 alignments of randomly picked orthologous sequences from *O. sativa* and *O. glaberrima* were used (grey line, see methods). Nucleotide substitution frequencies were calculated in a 400 bp sliding window with a 40 bp sliding step. (**b**) Insertion/Deletion (InDel) frequency calculated in a 1,000 bp sliding window with a 100 bp sliding step. (**c**) Proposed mechanism for error-prone DNA repair following the excision of DNA transposons. Step 1: after transposable element (TE) excision, 3′ overhangs are generated by exonuclease (blue). Step 2: the 3′ overhangs anneal using micro-homologies. To keep it simple, we only represent single-strand annealing (SSA[26,27]) here. Alternatively, the strands could also be connected via synthesis-dependent strand annealing (SDSA[26–28]), where the two strands are connected by 'filler' sequences (which were found in some cases, not shown). Step 3: new strands are synthesized by a replication complex that has deficiencies in DNA polymerases fidelity and mismatch repair. Step 4: the final repair product is rich in nucleotide substitutions and small insertions and deletions.

**TE insertions suggest repair patterns similar to excisions.**
Interestingly, we also found a slight increase in the number of
mutations close to TE insertion sites (Figs 2,3). The fact that we
observed this for DNA transposons as well as for retro-
transposons suggest that the underlying molecular mechanism
may be the same for both classes (Fig. 3). When
TEs insert into the genome, they produce a staggered cut with
5' overhangs[23]. The insertion therefore results in an intermediate
where the TE is ligated to short single-stranded segments, the
subsequent repair of which produces the TSD (Supplementary
Fig. 3). We propose that this intermediate can, in some cases,
become the target of 3'→5' exonucleases which expose longer
segments of single-stranded DNA (Fig. 3b). Repair of these

single-stranded stretches would then engage the same error-prone
replication complex as proposed for transposon excisions
(Fig. 2c). However, the proposed model would then also require
that TSDs themselves should, in many cases, not be perfect
repeats, but contain more substitutions than would be expected
from the overall mutation rate of the genome. We tested this
hypothesis by analysing insertion sites of 192 long terminal
repeats (LTRs) retrotransposons from three different families in
*O. sativa* (Supplementary Table 3). Due to their replication
mechanism, the two LTRs at the ends of the retrotransposon are
identical at the time of insertion. The 'age' of a retrotransposon
can therefore be estimated based on the differences the LTRs have
accumulated over time[24] (Supplementary Fig. 3b). By comparing
substitution rates in LTRs with those in TSDs, we found that
TSDs contain on average almost five times more substitutions
than LTRs (Supplementary Table 3). These data suggest that
second strand synthesis following a TE insertion is carried out by
the same error-prone replication complex as proposed for
excisions (Fig. 2), but that the single-stranded segments are on
average either shorter or produced only in rare cases.

**Excisions associate with elevated mutation rates in genes.**
Because DNA transposons preferably reside in gene promoters, we
expected that these regions should evolve at a particularly high
rate. Indeed, we found that the 2,000 bp upstream of genes
consistently contain 20–29% more nucleotide substitutions than
intergenic sequences from the same chromosomal region (Fig. 4;
Table 2). Because the genomes of the closely related *O. sativa* and
*O. glaberrima* are ~99.5% identical on average, the differences in
sequence conservation between promoters and intergenic
sequences are small, but the large sample size assures that they are
highly significant (*P* value <2.2E−16). Intergenic regions in rice
are mostly comprised of class 1 retrotransposons which are
believed to be largely free from selection pressure. It is therefore
intriguing that DNA repair following transposon excisions

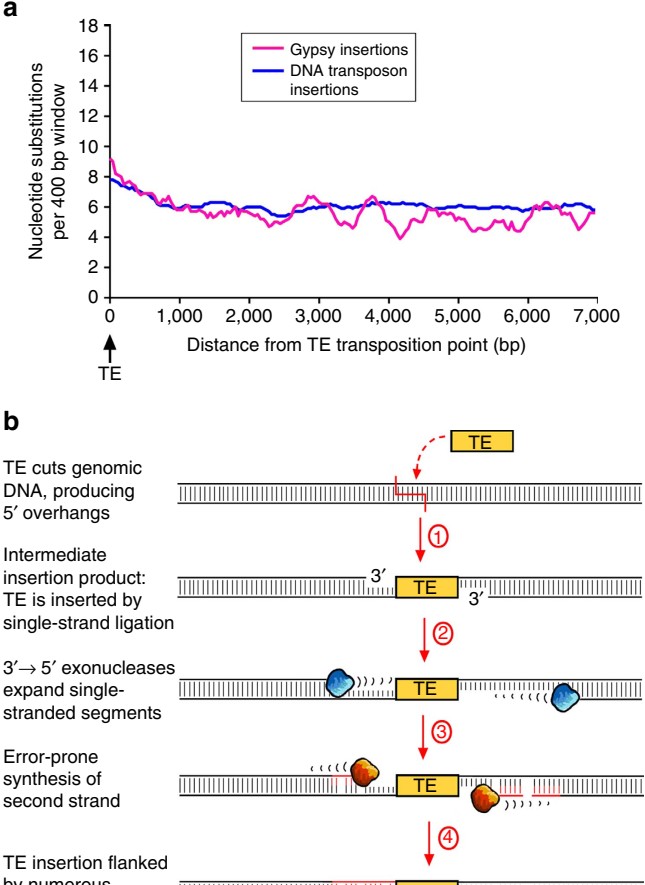

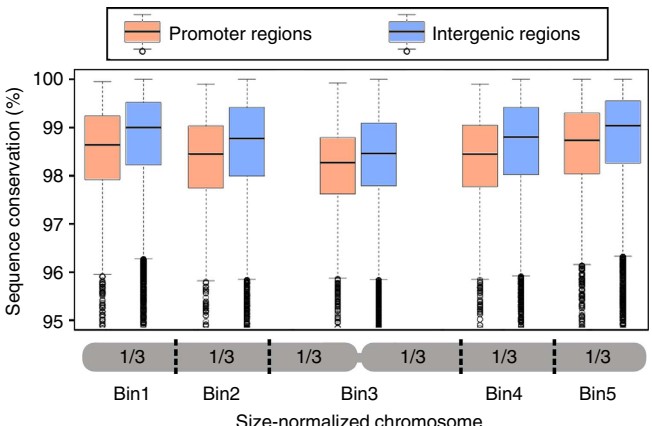

**Figure 3 | Mutations in sequences flanking transposable element insertions.** (**a**) Frequencies of nucleotide substitutions relative to insertion sites of DNA transposons and *Gypsy* retrotransposons sites in rice. In both cases, nucleotide substitution frequency increases slightly towards the insertion point. This indicates that insertions are also associated with small numbers of mutations in their flanking sequences. Furthermore, this result is evidence that events classified as DNA transposon insertions probably do not contain many precise excisions. (**b**) Proposed mechanism for error-prone DNA repair following TE insertions (see also Supplementary Fig. 3). Step 1: the TE inserts into the genome by producing a staggered cut, resulting in a TE that is ligated to the genomic DNA via single-stranded segments. Step 2: 3'→5' exonucleases expose large stretches of single-stranded DNA. Step 3: second strands are synthesized by a replication complex that has deficiencies in DNA polymerases fidelity and mismatch repair (the same as described in Fig. 2c). Step 4: the final TE insertion is flanked by segments rich in nucleotide substitutions and small insertions and deletions.

**Figure 4 | Sequence conservation along *Oryza sativa* and *Oryza glaberrima* chromosomes.** Data from all 12 chromosomes were compiled and chromosome sizes were normalized by dividing chromosome arms into three equally sized bins (*x*-axis). The *y*-axis depicts sequence identity of orthologous sequences. For each chromosome bin, promoter regions (the 2,000 bp upstream of the transcription start point, red box plots) are compared with intergenic sequences from the same bin (blue box plots). Promoters are on average 20–29% less conserved than intergenic sequences from the same chromosome bin. To calculate sequence conservation in intergenic regions, we isolated segments that are located in the middle of intergenic sequences which are at least 10 kb in size (that is, the distance between the end of one gene and the start of the next one is over 10 kb).

apparently leads to increased mutation rates of promoters to a degree that they evolve more rapidly than selectively neutral sequences. Interestingly, sequence conservation is generally lower in the centromeric and pericentromeric regions of chromosomes than in distal regions (Fig. 4), for which we have no explanation at this point.

The preference of DNA transposons to reside in up- and downstream regions of genes also implied that the 5′ and 3′ ends of coding sequences (CDS) should show an overall higher substitution rate than their central parts. Thus, we aligned CDS of closest homologues from *O. sativa* and *O. glaberrima* and studied overall sequence conservation as well as distributions of nucleotide substitutions along the aligned CDS. Overall, most CDS from *O. sativa* and *O. glaberrima* CDS are >99.5% identical. However, the distribution of sequence identities trails

off with some CDS being <97% identical (Supplementary Fig. 4). We expected that CDS which are >99.5% identical have not experienced transposon excisions in their vicinity, while genes with lower sequence identity could be those that have accumulated mutations due to a nearby transposon excisions. Indeed, we found that genes with lower than median sequence identity ranging from 98 to 99.4% show a >27% higher number of substitutions in their 5′ and 3′ regions than in the central part of the CDS (Fig. 5a; Supplementary Table 4), while genes with higher levels of sequence conservation do not show this pattern (Supplementary Fig. 5). Here, we only considered nucleotide substitutions in synonymous sites to exclude effects of differing selection pressures in different parts of the genes.

**SNP accumulations predict the presence of excision sites.** Since we predict that DNA repair following transposons excision is responsible for high numbers of mutations in their flanking sequences, regions containing above average numbers of mutations should, in turn, often contain transposon excision sites. Thus, we inspected sequence alignments from *O. sativa* and *O. glaberrima* that covered genes plus 3 kb of their flanking regions, and selected 50 segments that contained regions with local SNP accumulations ('high-SNP' set, examples in Supplementary Fig. 6). As a control, 50 segments with an overall low SNP density, similar to that of the genome-wide average were used (examples in Supplementary Fig. 6). The 100 alignments were manually searched in detail for the presence of polymorphic transposons and other insertions and deletions (InDels).

**Table 2 | Mean sequence conservation of promoter and intergenic sequences in different chromosome bins of *O. sativa* and *O. glaberrima*.**

| Chromosome bin | Promoter | Random | Difference* (%) |
|---|---|---|---|
| 1 | 98.22 | 98.62 | 28.99 |
| 2 | 98.03 | 98.35 | 19.39 |
| 3 | 97.8 | 98.17 | 20.22 |
| 4 | 98.06 | 98.39 | 20.50 |
| 5 | 98.33 | 98.58 | 17.61 |

*Difference in sequence divergence between promoter and intergenic (random) sequences.

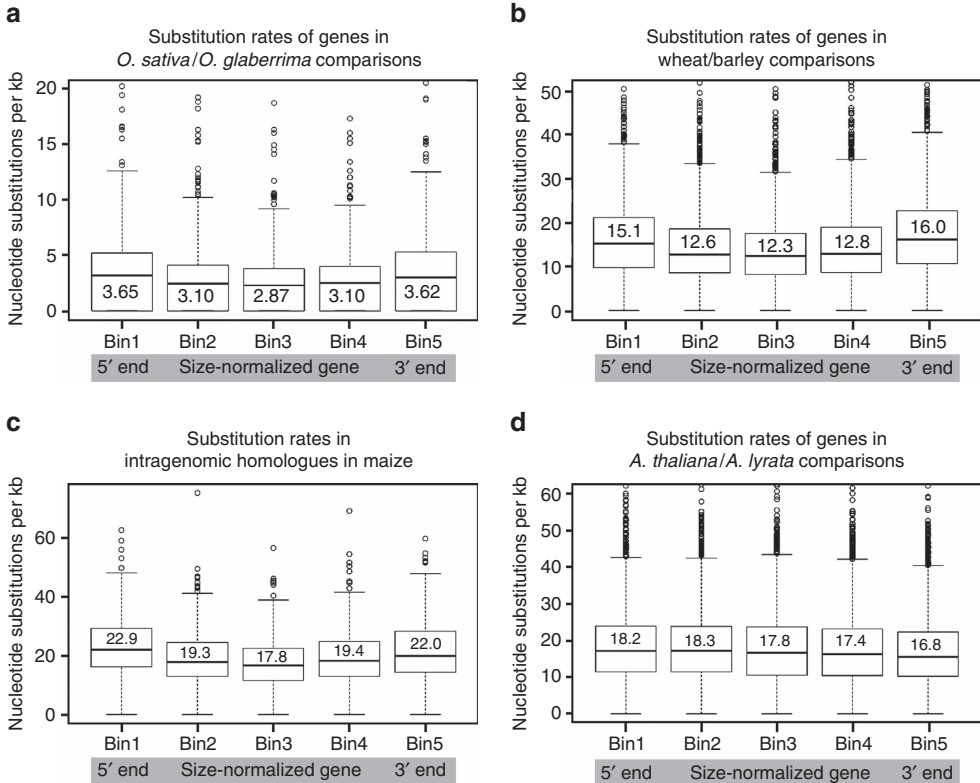

**Figure 5 | Substitution frequencies in synonymous sites showing that grass genes have higher mutation rates in their 5′ and 3′ regions.** To normalize the different CDS sizes, genes were divided into five equally sized bins and frequencies were normalized to nucleotide substitutions per kb for each bin. The bold line inside the box is the median value, while mean values are indicated with numbers. (**a**) Comparison of 442 closest homologues from *O. sativa* and *O. glaberrima*. (**b**) Comparison of 2,314 pairs of closest homologues from wheat and barley. (**c**) Comparison of 428 pairs of intragenomic closest homologues in maize that originate from a whole-genome duplication. (**d**) Comparison of 4,133 pairs of closest homologues from *A. thaliana* and *A. lyrata*.

**Table 3 | Test for predictability of presence of TE excisions based on SNP frequencies in *O. sativa* and *O. glaberrima*.**

| Polymorphism | Test set | Control set | *P* value |
|---|---|---|---|
| TE excisions | 16 | 2 | 0.0003 |
| TE insertions | 16 | 27 | 0.026 |
| Repeat slippage | 20 | 17 | 0.53 |
| InDel | 25 | 15 | 0.18 |

The sequences of the test set were chosen based on the presence of regions with high numbers of SNPs. Gene-containing regions that had a SNP density similar to that of the genome overall served as a control. Differences between test and control sets were tested with a $\chi^2$-test.

In the high-SNP data set, we identified 16 TE excisions, while in the control data set we only identified two excisions, a highly significant enrichment (Table 3). Interestingly, the high-SNP data set was also significantly depleted in transposons insertions, with 16 insertions identified in the high-SNP and 27 in the control data set (Table 3). This complements the above findings that transposon insertions only in rare cases are associated with SNP accumulations in their flanking regions (Fig. 3). We also surveyed InDels and repeat slippages (that is, differences in numbers of repeat units in micro- and minisatellites), since they can also result from DSB repair and thus could also be responsible of the introduction of SNPs. Here, we found no significant differences between the high-SNP and control data set. Although there are obviously several different causes for SNP accumulations, we identified transposon excisions as likely the main difference between regions that contain high numbers of SNPs and those which do not. Thus, these data show that local SNP accumulations can be used as search criterion for the identification of TE excisions.

**Increased mutation rates in genes are common in grasses.** Because all grass genomes sequenced so far are rich in DNA transposons, we predicted that we would find increased mutation rates also in genes from other grasses. We therefore compared closest gene homologues from wheat and barley, two species which diverged ~8 Myr ago[25]. Indeed, the 5′ and 3′ regions of the genes show a >20% higher number of substitutions than the central part of the genes (Fig. 5b). We also analysed maize where many genes are present in duplicates because maize is a relatively young polyploid that underwent a whole-genome duplication 5–10 Myr ago[2]. Thus, a comparison of such intragenomic closest homologues is analogous to a comparison of genes between two species. Here we found an even stronger effect, with 5′ and 3′ regions showing almost 30% more substitutions than the central part of genes (Fig. 5c). For both, the wheat/barley and the maize intragenomic CDS comparisons, the effects are statistically highly significantly (Supplementary Table 4). Considering that rice, maize, wheat and barley represent three different major clades of the grasses, our data strongly indicate that the described higher mutation rates in genes and regulatory sequences is common to all grasses. Interestingly, we did not find elevated mutation rates in genes in representatives of dicotyledonous plants ('dicots') such as *Arabidopsis*, *Brassica*, poplar and soybean (example in Fig. 5d; Supplementary Fig. 7; Supplementary Note 5). A *de novo* search for class 2 elements in these dicot genomes revealed that they contain at least 10–100 times fewer small DNA transposons than grasses (Supplementary Fig. 8; Supplementary Note 5). This result is in agreement with recent findings[26]. Furthermore, DNA transposons in *Arabidopsis* were found to be similarly active to those from rice[27,28], but their much lower numbers may diminish their impact, even if they have the same mutagenic effect per individual transposition event. Thus,

these data strengthen the correlation even more between the presence of DNA transposons and increased mutation rates of genes.

## Discussion

Data on how TEs contribute to gene evolution has been somewhat anecdotal (examples in refs 29–31). So far, most widely accepted is their role in altering gene expression. For example, a TE-mediated increase in expression level of the *tb1* gene in maize resulted in plants with fewer branches, a fundamental step in maize domestication[30,32]. We did indeed find that the presence of transposons is associated with higher levels of DNA methylation sites, suggesting an effect on transcription (Supplementary Fig. 9; Supplementary Note 6). However, the main contrast to previous studies is that our data show that transposon activity is associated with higher mutation rate and therefore may directly change coding sequences and regulatory regions by introducing nucleotide substitutions and InDels during DNA repair. We propose that error-prone repair of excision sites can introduce many mutations hundreds or even thousands of base pairs away from the sites. This would have the profound result that, even if the excision changes only a few base pairs at the actual transposon site[4,9,11], the entire genomic region accumulates mutations as a result of error-prone strand synthesis (Fig. 2; Supplementary Note 7). Most importantly, we show that this could affect thousands of genes in the species studied, and we provide evidence that this phenomenon is common to the vast family of the grasses with its over 10,000 species. Our data thereby also indicate that the highly successful types of non-autonomous DNA transposon elements that are associated with higher mutation rate and could therefore drive the accelerated evolution of genes only evolved after the separation of monocotyledons and dicotyledons ~145–300 Myr ago[33,34]. We previously showed that about 3% of the DNA transposons in rice have moved within the past 600,000 years, indicating that these elements are highly active[9]. Since DNA transposons are present in tens of thousands of copies in grasses[2,3], most genes will experience transposon excisions in their proximity at some point and therefore may accumulate particularly high numbers of mutations over time. Consequently, this may explain the stronger mutation rate gradient we found in more distantly related grasses such as wheat and barley (Fig. 5).

In plants and animals, a dominant DSB repair pathway is non-homologous end joining (NHEJ), where broken ends are directly joined, leading often to small deletions or insertions of 'filler' sequences[19,21]. Thus, NHEJ can explain certain repair patterns that were previously found at the immediate site of transposon excisions[4,9,11]. However, NHEJ does not require processing of the broken ends into single-stranded DNA. But our data strongly suggests that the repair pathway must involve single-stranded intermediates. Thus, our models are based on other known repair pathways. For this, we rely strongly on findings in yeast, where DNA repair processes are extremely well studied. We consider this legitimate, since most DSB repair pathways were probably established very early in eukaryote evolution. Indeed, practically all genes involved in DSB repair in yeast have homologues in plants, suggesting that DSB repair processes are virtually identical in plants and fungi[19–21]. Furthermore, studies on *Arabidopsis* mutants showed that many of these genes are involved in the same processes as in yeast[19]. For example, the yeast genes *Mre11*, *Rad50* and *Xrs2* which are required for micro-homology mediated end joining (the type on which our models are based) were shown to be involved in the same processes in *Arabidopsis*[20]. These findings are especially relevant for our model of DNA repair following TE insertions (which requires replication-independent 3′→5′ exonucleases for the extension of single-stranded regions), because the *Mre11* exonuclease

produces single-stranded DNA intermediates during DSB repair in yeast[17].

We propose that the activity of DNA transposons is a major driving force in the evolution of grasses, because DNA repair following transposon excisions may specifically accelerate evolution of genes. Our findings may, in part, explain the phenomenal evolutionary success of the grasses, a very large group of plants that contains the most important crops such as rice, maize, wheat, sorghum and barley which are the basis of most food consumed by humankind.

## Methods

**Survey of DNA transposon distribution relative to rice genes.** A total of 101 sequences of *DTT_Mariner* and *DTH_Harbinger* transposons from rice were obtained from the TREP database (wheat.pw.usda.gov/ITMI/Repeats/). They represent 19 *DTT_Mariner* and 25 *DTH_Harbinger* families. The 101 sequences were mapped with blastn to the *O. sativa* genome (version 6) using an in-house Perl script. The cutoff for blast hits was 50 bp and 80% sequence identity. If multiple TE families mapped to the same location, the one with the strongest blastn hit was chosen. To analyse their position relative to genes, the TE annotation was then cross-matched with the gff format gene annotation of the rice genome. We used the annotated transcription start and end points as anchor points and generated a data set of the positions of all annotated TEs within 5 kb upstream of the transcription start point and 5 kb downstream of the transcription end point for each gene. Furthermore, positions of TEs inside the gene were recorded. We selected genes larger than 4 kb and recorded TE positions within 2 kb from each end of a gene. For simplicity, only genes in forward orientation were used. The final dataset included data for 4,994 genes. Sequences covered by TEs were added up for all genes, resulting in a final coverage plot that reflects the overall distribution of TEs relative to genes (Supplementary Fig. 1).

**Identification of transposon polymorphisms.** We used an alignment of ~60% of the genomes of *O. sativa* and *O. glaberrima* described in our previous study[9] to identify insertions larger than 50 bp. Insertions were screened for homology with TE sequences by blastn against the TREP database (wheat.pw.usda.gov/ITMI/Repeats/). Using an in-house Perl script, TEs with the highest homology were mapped onto the *O. sativa*/*O. glaberrima* alignments to facilitate visual inspection and to classify the polymorphism as transposon insertion or excision. Over 2,000 polymorphisms were screened, yielding the 482 insertions and 158 excisions (Table 1; Supplementary Tables 1 and 2).

**Test for orthology of the analysed loci.** To ensure that the aligned sequences from *O. sativa* and *O. glaberrima* indeed come from orthologous loci, we mapped the sequences used for the alignments back onto both genomes. That is, the sequences from *O. sativa* were first mapped back to the *O. sativa* genome and then mapped on to the *O. glaberrima* genome. The same was done *vice versa* with the corresponding *O. glaberrima* sequence. We split the aligned 24 kb regions into segments of 1,000 bp and mapped each segment by blastn to the genome it came from as well as to the genome of the other species. This was done because blast alignments are often fragmented due to the presence of low-complexity sequences or TE insertions in one or the other species. Therefore, one cannot expect a long sequence from one species to produce a similarly long blast hit in another. We therefore rather assigned each locus a score for how many of the segments map in the putative orthologous region in the other genome as a quantitative assessment of how strong the evidence for true orthology is for a particular locus.

For each 1,000 bp segment, we recorded the positions of the top blast hit in the genome it came from as well as to the genome of the other species. We required that the top blast hit produced an alignment of at least 600 bp. Thus, some segments could not be mapped due to the presence of low-complexity sequences that are filtered out in the blastn search. Furthermore, one expects that not all segments map unambiguously to the orthologous locus in the other genome. This can, for example, be due to a large retrotransposon insertion in one species. The segments covering that retrotransposon would have no counterpart in the orthologous locus in the other species and therefore map elsewhere in the genome. The genomic region where the majority of the segments map was considered the putative ortholog. Furthermore, since we ran the analysis in both directions, we required that sequences from both species had to identify each other as the closest homologue. All analysed loci fulfilled these criteria. Additionally, as Supplementary Fig. 2 shows, all except two loci are located in perfect colinear order along the chromosomes.

**Distinguishing transposon insertions and excisions.** We defined a TE polymorphism as an insertion if one species contained the TE plus the duplicated target site (TSD) on both sides, while the other species only contained one copy of the target site. Excision are more difficult to define as they can go along with various re-arrangements[9,11]. In general, we defined an excision by the absence of the TE in one species, with the pattern differing from that of an insertion. We

distinguished different types of excisions: (i) in a perfect excision, as previously defined[11], one species contains the TE with flanked by the two units of the TSD while the other species does not contain the TE but both copies of the TSD. (ii) Excisions with deletions were defined as the TE plus some flanking sequences being absent in one species. To distinguish these events from random deletions that by chance removed the TE plus flanking regions, we requested that one breakpoint of the excision be within 3 bp of one end of the TE (we considered it unlikely that a random deletion would have one of its borders so close to the end of a TE). (iii) Excisions with fillers were defined as events where the TE in one species is replaced with a completely unrelated sequence in the other. Fillers can range from a few bp to several kb. Also here, we requested that the end of the filler sequence be within 3 bp of one end of the TE. Filler insertions were often found combined with deletions as described in (ii). Additional methodological considerations on distinguishing transposon excisions from insertions are provided in Supplementary Note 3.

**Quantification of mutations flanking polymorphic TEs.** For all identified insertions and excisions, 12 kb of the flanking sequences were extracted from the *O. sativa* and *O. glaberrima* genome-wide alignment. We selected all alignments where >7,000 bases could be aligned (due to large insertions and deletions and/or colinearity breaks, usually <12 kb were actually aligned). This selection resulted in 206 sequence alignments for excisions and 438 for insertions. The transposon excision/insertion site was used as anchor point (that is, position zero) from which all nucleotide substitutions and InDels were recorded. Sequence polymorphisms were added up for all alignments relative to the TE excision/insertion site. For the graphical representation (Fig. 2a,b), nucleotide substitutions and InDel densities were calculated by a running average.

**Survey of LTR retrotransposon insertions.** Consensus sequences of LTRs from the *O. sativa* retrotransposon families *RLG_Cara*, *RLG_Houba* and *RLG_hopi* were used in blastn searches against the *O. sativa* genome. LTRs of the same family wich were found in the same direction and <14 kb apart were considered candidates for full-length elements. These including the 5 bp flanking sequences (corresponding to the TSD) were extracted from the genome. All candidate elements were visually inspected by DotPlot against a reference sequence of the respective retrotransposon family, to ensure that indeed full-length elements were selected (instead of, for example, two solo-LTRs that just happen to be located near each other). All LTR pairs of the individual copies were aligned with the programme WATER (emboss package, emboss.sourceforge.net/) to determine the number of substitutions between LTRs. From this, the average sequence conservation of the LTRs for each retrotransposon family was calculated (we excluded LTR pairs where sequence homology was over two standard deviations lower than that of the entire family, since such events could be results inter-element recombination). Analogously, the TSD sequences of all copies were aligned. The total number of mismatches in TSDs was then compared to that in LTRs. A $\chi^2$-test was used to test if the two values differed from each other (Supplementary Table 3).

**Comparison of promoters from O. sativa and O. glaberrima.** Information on start and end point of genes was extracted from the gff format annotation of the rice genome. As start and end point of genes we used transcription start and end points. Here, we used rice genome version 5, because our previously published genome alignment of *O. sativa* and *O. glaberrima*[9] was done with this version. We defined the region from the transcription start point to 2 kb upstream of it as promoter region. Alignments were accepted when >600 bp in this 2 kb region could be aligned between *O. sativa* and *O. glaberrima*. For comparison, alignments of intergenic sequences were used. Here, we isolated segments that are located in the middle of intergenic sequence that are at least 10 kb in size (that is, the distance between the end of one gene and the start of the next one is over 10 kb). Because sequence conservation along chromosome varies (Fig. 3), chromosome arms were divided into three equally sized bins for comparison of promoter and intergenic sequences. Data for promoters and intergenic sequences were analysed separately for each chromosome bin. To test whether the data sets for the individual bins differ from each other, the wilcox.test programme from RStudio (rstudio.com) was used.

**Comparison of CDS of genes.** Repositories where CDS of different species were obtained are listed in Supplementary Table 5. CDS for *O. glaberrima* were deduced from aliment with *O. sativa* CDS and are available upon request. Closest homologues from different species or, in the case of maize, homeologs that originated from a whole-genome duplication were identified by bi-directional blastn searches. Only homologues which had each other as the top blastn hit were used for comparison. Bi-directional closest homologues were aligned at the protein level using the programme WATER from the EMBOSS package (emboss.sourceforge.net). The aligned protein sequences were back-translated to ensure that corresponding codons were aligned. We considered only alignment positions corresponding to the third codon base for Ala, Gly, Leu, Pro, Arg, Ser, Thr and Val. For those amino acids which all have six possible codons (Leu, Arg and Ser), we used only the codons starting with CT, TC and CG, respectively (that is, the codons in which the third base can be exchanged without causing an amino acid change). To normalize the different sizes of genes, the aligned CDS were split into five equally sized bins. To obtain sufficiently high numbers of synonymous substitutions, we used only gene pairs where

> 1,500 bp of the CDS could be aligned. For each bin of each gene, we calculated the number of synonymous substitutions per kb. Finally, we compiled the data for the five bins for all genes. To test whether the data sets for the individual bins differ from each other, the wilcox.test programme from RStudio was used.

**De novo identification of small DNA transposons in dicots.** DNA transposons are characterized by the presence of terminal inverted repeats which serve as binding site for transposase enzymes[35]. The initial step of de novo identification was to screen chromosomal segments in windows of 1,000 bp, which overlap by 500 bp. The 1,000 windows were aligned with the programme WATER from the EMBOSS package against themselves in reverse orientation. Outputs were parsed and visually inspected for the presence of inverted repeats longer than ∼15 bp and over ∼70% identity. The candidate sequences (inverted repeat and the sequences between them) were excised from the 1,000 bp. The candidate TEs were then used in blastn searches against the respective genome. Sequences with multiple hits were considered true DNA transposons. The de novo detection was done on one entire Arabidopsis chromosome, 2 Mbp of poplar linkage group 1 and 500 kb of rice chromosome 10 (Supplementary Fig. 8).

**Comparative analysis of DNA methylation.** Data on methylation sites in O. sativa and O. glaberrima were kindly provided by Detlef Weigel and Claude Becker (Max Planck Institute for Developmental Biology, Tübingen, Germany). These data sets will be published elsewhere (personal communication, Detlef Weigel and Claude Becker). Sequence segments of 4 kb spanning the polymorphic transposon in O. sativa and O. glaberrima were extracted from the chromosomes. Methylated sites were flagged and the sequence segments were aligned with the programme Water (emboss package, emboss.sourceforge.net/). Since we found that practically no methylation sites were conserved between the two species, methylation states were compared by simply counting the numbers of methylated sites in the sequences segments from the two species. The ratio of the number of methylation sites in O. sativa and O. glaberrima was then calculated for each transposon locus. For comparison, a second segment 2,000–4,000 bp downstream of the transposon was extracted.

**Statistics.** Wilcoxon rank sum test was used to test whether substitution rates in different bins of size-normalized genes differ from each other. Sample sizes depended on how many bi-directional closest homologues could be identified between species. Sample sizes are provided in Fig. 5. To test if SNP accumulations can be used to predict transposon excisions, results of 50 candidate sequences were compared with those of 50 control sequences. The sample size of 50 was used to meet the commonly used small sample size criteria. A $\chi^2$-test was used to test for differences between test and control sets. To test if substitution rates in TSDs differ from those in LTRs, 192 full-length LTR retrotransposons were isolated from the rice genome. Sample size was determined by the copy number of retrotransposons. A $\chi^2$-test was used to test for differences between substitution rates in TSDs and LTRs.

**Data availability.** Repositories where CDS of different species were obtained are listed in Supplementary Table 5. The genome sequence of O. glaberrima can be obtained from Gramene (ensembl.gramene.org). The authors declare that all other data supporting the findings of this study are available within the manuscript and its Supplementary Information Files or are available from the corresponding author upon request (such as original software and sequence aliments of genomic and CDS sequences).

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

## Acknowledgements

We thank Detlef Weigel and Claude Becker for providing methylome data for O. sativa and O. glaberrima. This material is based on work supported by the Swiss National Foundation grant #31003A_138505/1 to T.W., by the US National Science Foundation under grants #0321678, #0638541, #0822284 and #1026200 to Y.Y., and R.A.W. and the Bud Antle

Endowed Chair of Excellence in Agriculture and Life Sciences and the AXA Endowed Chair of Genome Biology and Evolutionary Genomics to R.A.W. Any opinions, findings and conclusions or recommendations expressed in this material are those of the authors and do not necessarily reflect the views of the US National Science Foundation.

## Author contributions

T.W. designed the study, analysed the data and wrote the paper. S.R. helped design the study, created software, analysed the data and provided critical input in writing the manuscript. R.A.W., Y.Y., S.R., M.C., P.R.M. and A.Z. produced the genome sequence. K.F.X.M., G.H. and O.P. produced the genome annotation.

## Additional information

**Competing financial interests:** The authors declare no competing financial interests.

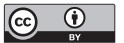

