## [Peer Review File · Nature Communications]

Reviewers' comments:

Reviewer #1 (Remarks to the Author):

The manuscript describes the studies of mutations resulted from DNA transposon transposition events in grass species. The authors report that DNA sequences close to a transposon site particularly excision site has significant more sequence mutations. A model of error prone DNA replication following double stranded break was used to explain the observation. The report analyzed the mutation rates of 5', 3' and central section of genes and found that the ends have higher mutation rates for the genes with relatively higher sequence divergence. The authors hypothesized that such increased mutation rates at the ends may be caused by transposition events.

The studies and reported results are novel and very interesting. In fact, in my own experience of analyzing polymorphisms resulted from DNA element transposition, I sometimes had a gut feeling that the sequences next to the polymorphic sites have increased mutation rates, it is exciting to see that the authors are able to systematically address this aspect and make the remarkable findings. The statistics are appropriate and treatment of uncertainties is reasonable. The conclusions are based on robust analyses and are convincing. The manuscript is well written and easy to follow.

Suggestion for improvement:

The authors demonstrated the increased mutation rates at ends of genes with relatively high sequence divergence and proposed that this observation is likely to have resulted from transposition events. Among those above median divergence levels (those showed increased mutation rates at ends of genes), sequence alignments may be able to reveal footprints of transposition events. If excision footprints or insertion of an element can be found close to these genes, it further strengthens the point.

Reviewer #2 (Remarks to the Author):

I read the short manuscript of Wicker et al with great interest. Using bioinformatic tools the authors demonstrate that DNA transposons have a significant influence on the evolution of the genomes of monocots. By comparative analysis of Oriza species they demonstrate that there is an increase in mutations close to transposon excision sites. This has the consequence that regions in the genome with more transposons have higher mutations rates. Thus mutation rate are increased especilly close to genes. This not only applies for rice but also for other monocots like maize or barley but not for dicots which have much less transposon activity than monocots.

The authors speculate that the reason for the increased mutation rate is due the involvement of error-prone DNA polymerases complexes in fill-in repair reactions in plants like in yeast, which seems a likely hypothesis to me

This is an interesting paper that well deserves a wide audience.

Reviewer #3 (Remarks to the Author):

In an analysis focused on sequence data from the *Oryza* genus and supported with data from barley, wheat, and maize, the authors show that the regions flanking excision sites for DNA transposons contain over 10-fold more SNPs and indels than the genome as a whole. The regions extend up to 3 kb from the sites, and the mutations are particularly plentiful in the upstream regions of genes, which contain promoters and are hotspots for DNA transposon insertions. A mechanism is proposed for the formation of these mutations. The data are novel, in that repair of the DSBs induced by TE insertions were earlier viewed as either perfect, generating the expected tandem target site duplications, or involving only short mismatched repair regions. Wicker et al show that these are extensive in number and size, and not confined to rice. The results are of widespread interest to the plant community due to their implications for promoter regions and thereby gene function and for plant genome evolution. The ever-increasing genome-resequencing efforts will no doubt show that the phenomenon is general, at least for the grasses. The lack of the phenomenon in dicots is correlated with the lower abundance of the Class II elements there. The methods were straightforward and adequate to answer the question.

My only criticism with the need for spelling and grammatical corrections: the multiple occurrence of "barely" instead of "barley"; "encoded my a small number of "mother" elements"; "transposons excisions"; "transposons loci", to give a few examples.

Reviewer #4 (Remarks to the Author):

This manuscript reports an interesting finding of elevated mutation rates around transposon excision/insertion sites in rice. Authors discuss the potential evolutionary role of enhanced sequence diversification, especially in promoter regions, in grasses. Although the observations are certainly interesting, authors are encouraged to provide a deeper discussion on their data.

1) Based on the model, I understand elevated mutation rates around excision sites. But how would this model explain mutations around insertions? Strand resection is certainly not expected to occur during transposon insertion; yet, higher sequence polymorphism is also observed around insertions. Why?

2) Fig. 2A/B show nucleotide substitutions as far as 7 kb away from excision/insertion sites. How can this be possibly explained? Based on the model, the effect is expected to be strictly local.

3) Authors propose the involvement of error-prone processes in DSB repair after transposon excision, including DNA polymerase zeta, or DSB-induced complex with reduced fidelity. I assume these are not the only mechanisms in grasses for the repair of DSBs. Why would

these be so preferentially be used over any other mechanism?

4) Authors refer to processes in yeast. I think it would be useful to discuss DSB repair processes in plants in order to explain the observations. In other systems, nonhomologous end joining has been shown to be primarily involved in the repair of DSBs generated by transposon excision. However, NHEJ is not expected to generate the sequence polymorphisms observed here.

5) Authors argue that dicots have fewer transposons than grasses, and perhaps that is why sequence divergence at the 5' and 3' ends of genes is not so substantial as in grasses. However, the mutagenic effect of a transposon excision must be the same on a per locus basis, assuming DSB repair following transposon excision is similar in dicots and grasses. Again, a discussion of DSB repair processes in plants would be useful.

General information about response to the reviewers' comments:

1. We provide a PDF file of the manuscript where changes that were made in response to the reviewers' comments are highlighted (file title: manuscript_changes_highlighted.pdf). We refer to this file, (Page and paragraph numbers), when we describe the changes that were made. The file also has the figures embedded for a more easy read.
2. Since we have expanded the manuscript by adding three paragraphs and a few additional statements, we decided to add subtitles to the results section to give the manuscript a structure that is easier to follow.
3. We removed two redundant sentences from the discussion (Page 9, 2nd paragraph).
4. Changes that were made to conform with the journals style and policies are not highlighted in the PDF.

Point-by-point response to the reviewers' comments:

Reviewers' comments:

Reviewer #1 (Remarks to the Author):

The manuscript describes the studies of mutations resulted from DNA transposon transposition events in grass species. The authors report that DNA sequences close to a transposon site particularly excision site has significant more sequence mutations. A model of error prone DNA replication following double stranded break was used to explain the observation. The report analyzed the mutation rates of 5', 3' and central section of genes and found that the ends have higher mutation rates for the genes with relatively higher sequence divergence. The authors hypothesized that such increased mutation rates at the ends may be caused by transposition events.

The studies and reported results are novel and very interesting. In fact, in my own experience of analyzing polymorphisms resulted from DNA element transposition, I sometimes had a gut feeling that the sequences next to the polymorphic sites have increased mutation rates, it is exciting to see that the authors are able to systematically address this aspect and make the remarkable findings. The statistics are appropriate and treatment of uncertainties is reasonable. The conclusions are based on robust analyses and are convincing. The manuscript is well written and easy to follow.

Suggestion for improvement:

Reviewer's comment:

The authors demonstrated the increased mutation rates at ends of genes with relatively high sequence divergence and proposed that this observation is likely to have resulted from transposition events. Among those above median divergence levels (those showed increased mutation rates at ends of genes), sequence alignments may be able to reveal footprints of transposition events. If excision footprints or insertion of an element can be found close to these genes, it further strengthens the point.

Our response:

This is an excellent suggestion. We addressed this comment by manually analyzing in detail 50 sequence alignments of *O. sativa* and *O. glaberrima* which contain genes with high numbers of SNPs. As control, we used gene-containing regions that did not show SNP accumulations. Indeed, we found that the presence of local SNP accumulations can be used as a predictor of transposon excisions (16 in the test set vs. 2 in the control set, a highly significant enrichment). Not surprisingly,

we also found several SNP-rich regions without transposons excisions, since there may be many other sources of DSBs that are followed by error-prone repair. Nevertheless, this analysis showed a very strong association between SNP accumulations and transposon excisions. We added a section to the results (Page 7, 2nd paragraph) plus a table (Table 3) and a supplementary Figure (Supplementary Figure 6.) describing this analysis.

Reviewer #2 (Remarks to the Author):

I read the short manuscript of Wicker et al with great interest. Using bioinformatic tools the authors demonstrate that DNA transposons have a significant influence on the evolution of the genomes of monocots. By comparative analysis of *Oryza* species they demonstrate that there is an increase in mutations close to transposon excision sites. This has the consequence that regions in the genome with more transposons have higher mutations rates. Thus mutation rate are increased especially close to genes. This not only applies for rice but also for other monocots like maize or barley but not for dicots which have much less transposon activity than monocots.

The authors speculate that the reason for the increased mutation rate is due the involvement of error-prone DNA polymerases complexes in fill-in repair reactions in plants like in yeast, which seems a likely hypothesis to me

This is an interesting paper that well deserves a wide audience.

Our response:

We very much appreciate the positive feedback.

Reviewer #3 (Remarks to the Author):

In an analysis focused on sequence data from the *Oryza* genus and supported with data from barley, wheat, and maize, the authors show that the regions flanking excision sites for DNA transposons contain over 10-fold more SNPs and indels than the genome as the whole. The regions extend up to 3 kb from the sites, and the mutations are particularly plentiful in the upstream regions of genes, which contain promoters and are hotspots for DNA transposon insertions. A mechanism is proposed for the formation of these mutations.

The data are novel, in that repair of the DSBs induced by TE insertions were earlier viewed as either perfect, generating the expected tandem target site duplications, or involving only short mismatched repair regions. Wicker et al show that these are extensive in number and size, and not confined to rice. The results are of widespread interest to the plant community due to their implications for promoter regions and thereby gene function and for plant genome evolution. The ever-increasing genome-resequencing efforts will no doubt show that the phenomenon is general, at least for the grasses. The lack of the phenomenon in dicots is correlated with the lower abundance of the Class II elements there. The methods were straightforward and adequate to answer the question.

Reviewer's comment:

My only criticism with the need for spelling and grammatical corrections: the multiple occurrence of "barely" instead of "barley"; "encoded my a small number of "mother" elements"; "transposons excisions"; "transposons loci", to give a few examples.

Our response:

We carefully proofread the manuscript and corrected spelling and grammatical errors.

Reviewer #4 (Remarks to the Author):

This manuscript reports an interesting finding of elevated mutation rates around transposon excision/insertion sites in rice. Authors discuss the potential evolutionary role of enhanced sequence diversification, especially in promoter regions, in grasses. Although the observations are certainly interesting, authors are encouraged to provide a deeper discussion on their data.

Reviewer's comment:

1) Based on the model, I understand elevated mutation rates around excision sites. But how would this model explain mutations around insertions? Strand resection is certainly not expected to occur during transposon insertion; yet, higher sequence polymorphism is also observed around insertions. Why?

Our response:

We include now an additional model that explains this (Figure 3) and we support it by citing studies that describe the types of exonuclease that could be involved in the process. Furthermore, we performed an analysis of 192 LTR retrotransposon insertions in rice. The "age" of retrotransposons can be estimated by comparing sequence divergence of their long terminal repeats (LTRs). If error-prone DSB repair plays a role, the (usually) 5 bp target site duplications should overall show a higher mutation rate than the "aging" retrotransposons. Indeed, we found that the TSDs contain approximately 3-10 times more mutations than would be expected from LTR divergence. These data provide strong empirical support for the proposed model. Therefore, we dedicated a section of the main manuscript (Page 5, 2nd paragraph) and a figure (Figure 3, Page 23) to this analysis (before, we only referred to a Supplementary Figure). We also added a brief section to the methods (Page 14, 2nd paragraph) and additional explanations in Supplementary Figure 3.

Reviewer's comment

2) Fig. 2A/B show nucleotide substitutions as far as 7 kb away from excision/insertion sites. How can this be possibly explained? Based on the model, the effect is expected to be strictly local.

Our response:

There are multiple studies that show that the exposed 3' overhangs can reach sizes of thousands of bp in yeast. And there are several studies indicating that this is similar in plants, due to the high conservation of DSB repair pathways (and which also exactly fits our the observations). We emphasize this point now more strongly and have added more references to respective studies (Page 5, 1st paragraph). See also our answer to the reviewer's comment 4 and 5 below.

Reviewer's comment:

3) Authors propose the involvement of error-prone processes in DSB repair after transposon excision, including DNA polymerase zeta, or DSB-induced complex with reduced fidelity. I assume these are not the only mechanisms in grasses for the repair of DSBs. Why would these be so preferentially be used over any other mechanism?

Our response:

We mention DNA polymerase zeta specifically because it was shown to be involved in translesion synthesis in yeast that results in accumulation of SNPs in the region of the DSB, an outcome that is practically identical to what we observed for transposon excisions. To address the reviewer's comment, we state now clearly that a plant homolog of DNA polymerase zeta is one possible enzyme that could be involved, and we also specifically refer to the publication that showed it to be error-prone (Page 5, 1st paragraph).

Reviewer's comment:

4) Authors refer to processes in yeast. I think it would be useful to discuss DSB repair processes in plants in order to explain the observations. In other systems, nonhomologous end joining has been shown to be primarily involved in the repair of DSBs generated by transposon excision. However, NHEJ is not expected to generate the sequence polymorphisms observed here.

Our response:

We agree that DSB repair in plants should be discussed in more detail. We have therefore added a paragraph to the discussion that reviews studies on DSB repair in plants. We cite now multiple studies that provide strong evidence that DSB repair pathways are extremely similar in plants and fungi (Page 10, 2nd paragraph). We also cite them now in the paragraph where the molecular model is proposed (Page 4, 3rd paragraph). Additionally, we also emphasize clearly in the discussion that our models are not based on NHEJ, because NHEJ does not involve processing of the broken ends into single-stranded DNA (which is why error-prone second strand synthesis would not come into play at all, Page 10, 2nd paragraph). We also added the distinction between NHEJ and those that involve single-stranded intermediates to the introduction (Page 3, 2nd paragraph) and Supplementary Note 1 (1st paragraph).

Reviewer's comment:

5) Authors argue that dicots have fewer transposons than grasses, and perhaps that is why sequence divergence at the 5' and 3' ends of genes is not so substantial as in grasses. However, the mutagenic effect of a transposon excision must be the same on a per locus basis, assuming DSB repair following transposon excision is similar in dicots and grasses. Again, a discussion of DSB repair processes in plants would be useful.

Our response:

In addition to the discussion on DSB repair in plants (see comment above), we have also added a statement with references to TE activity in Arabidopsis and Brassica. It states that comparative studies indicated that DNA transposons are active at similar levels to those in rice but their overall low number result in much approximately 10-100 times fewer excisions and insertions. Thus, even if the mutagenic effect on a per-locus (or per-transposition) basis is the same, the impact on mutation rates would be minimal (Page 9, 1st paragraph).

REVIEWERS' COMMENTS:

Reviewer #1 (Remarks to the Author):

The revised manuscript fully addressed my comments and the additional analyses strengthened the major conclusions.

Reviewer #4 (Remarks to the Author):

Authors have addressed my comments in a satisfactory manner.